# Differential Sensitivity of Melanoma Cells and Their Non-Cancerous Counterpart to Cold Atmospheric Plasma-Induced Reactive Oxygen and Nitrogen Species

**DOI:** 10.3390/ijms232214092

**Published:** 2022-11-15

**Authors:** Sun-Ja Kim, Min-Jeong Seong, Jong-Jin Mun, Jin-Hee Bae, Hea-Min Joh, Tae-Hun Chung

**Affiliations:** Department of Physics, Dong-A University, Busan 49315, Republic of Korea

**Keywords:** cold atmospheric pressure plasma jet, plasma-cell interaction, reactive oxygen and nitrogen species, plasma selectivity

## Abstract

Despite continuous progress in therapy, melanoma is one of the most aggressive and malignant human tumors, often relapsing and metastasizing to almost all organs. Cold atmospheric plasma (CAP) is a novel anticancer tool that utilizes abundant reactive oxygen and nitrogen species (RONS) being deposited on the target cells and tissues. CAP-induced differential effects between non-cancerous and cancer cells were comparatively examined. Melanoma and non-cancerous skin fibroblast cells (counterparts; both cell types were isolated from the same patient) were used for plasma–cell interactions. The production of intracellular RONS, such as nitric oxide (NO), hydroxyl radical (•OH), and hydrogen peroxide (H_2_O_2_), increased remarkably only in melanoma cancer cells. It was observed that cancer cells morphed from spread to round cell shapes after plasma exposure, suggesting that they were more affected than non-cancerous cells in the same plasma condition. Immediately after both cell types were treated with plasma, there were no differences in the amount of extracellular H_2_O_2_ production, while Hanks’ balanced salt solution-containing cancer cells had lower concentrations of H_2_O_2_ than that of non-cancerous cells at 1 h after treatment. The melanoma cells seemed to respond to CAP treatment with a greater rise in RONS and a higher consumption rate of H_2_O_2_ than homologous non-cancerous cells. These results suggest that differential sensitivities of non-cancerous skin and melanoma cells to CAP-induced RONS can enable the applicability of CAP in anticancer therapy.

## 1. Introduction

The biomedical application of cold atmospheric plasma (CAP) has been regarded as one of the most effective approaches in plasma science [1,2,3,4,5,6,7,8,9,10,11,12,13]. The field of plasma medicine is advancing rapidly toward the development of new medical therapies, including dental care, dermatology, and cancer therapy [14,15,16]. The eradication of cancer cells by CAP is an alternative to conventional treatments, such as surgery, chemotherapy, and radiation therapy, and has emerged as a novel treatment strategy [9,17,18]. The CAP-generated reactive oxygen and nitrogen species (RONS) are considered the driving force behind CAP-induced cellular responses, such as decreased proliferation, cell cycle arrest, and cell death in cancer cells [17,19,20]. In indirect CAP treatment, the cell culture medium or buffer solution was first exposed to CAP treatment and then added to medium-free cells. Saadati et al. found a significant cell death and substantial reduction in tumor growth in direct CAP treatment compared to indirect CAP treatment [21].

RONS are considered to be normal byproducts of numerous cellular processes. Low levels of RONS act as signaling molecules that facilitate cell survival, namely, adequate levels of RONS are important for cell homeostasis involved in the development of cellular processes [22]. In contrast, high levels of RONS can have anticancer properties, including increasing oxidative stress, damaging DNA, and inducing apoptosis [23,24]. Despite decades of research on plasma cancer therapies, there is still widespread concern about selectivity between cancer and non-cancerous cells. Some claimed that non-cancerous cells are also dose-dependently damaged by plasma exposure [1,25]. Thus, it is important to investigate the precise effects of specific RNS and ROS, especially for selectivity toward cancer therapy. Superoxide (O_2_^−^), hydrogen peroxide (H_2_O_2_), nitric oxide (NO), and the hydroxyl radical (•OH) are among several types of RONS generated by plasma, and their important influence on cellular chemistry is well known [26,27]. Bauer et al. concluded that relatively high concentrations of CAP-derived NO, to some extent, contribute to CAP-mediated apoptosis induction in tumor cells through triggering tumor cell-specific extracellular singlet oxygen formation, and singlet oxygen has a prominent role during the inactivation of catalase [26]. While the exact mechanism remains unclear, there is a general consensus that points to these RONS produced by CAP being a major factor in the instigation of apoptosis in cancer cells [1,18]. •OH is one of the most active species generated in moist gas mixtures, and it is so reactive that it is thought to have dramatic intracellular effects [28]. H_2_O_2_ is generated by various reaction pathways, including reactions between •OH species [29]. H_2_O_2_ has well-known biological activities, including the induction of DNA damage and a role in mitogenic stimulation and cell cycle regulation [30]. Persistently upregulated H_2_O_2_-dependent signaling pathways are involved in cell growth and survival in many cancer cells, yet high levels of H_2_O_2_ can also induce cell cycle arrest or apoptosis [31]. An increase in the cellular levels of H_2_O_2_ can sensitize cancer cells selectively to H_2_O_2_-induced cell death [31,32]. Cancer cells under increased oxidative stress are likely to be more vulnerable to damage by further RONS induced by exogenous stimuli due to an impaired redox balance [31]. Because of this functionality, the appropriate application of RONS produced by CAP can be a potential tool for anticancer treatment [16].

In our previous studies [33,34], a remarkable overproduction of RONS and reduction in cell viability were observed when cancer cells were exposed to CAP, while non-cancerous cells were less affected by the CAP-mediated RONS, and cell viability was less changed. Biscop et al. asserted that definitive proof of selectivity is scarce because of the discrepancies between treatment conditions for cancerous and non-cancerous cells. They also pointed out that a selectivity study of CAP treatment should be performed with cancerous and non-cancerous cells from the same tissue, same cell type, and cultured in the same medium [8]. They reiterated that a comparison should be made between equivalent cell types before the selectivity of CAP treatment can be claimed. Although previous studies observed plasma-induced intracellular RONS to increase in cancer cells [19,34], it remains unclear which one among these species plays a crucial role in the selectivity of non-cancerous and cancer cells. The intracellular response to plasma should be carefully examined in various types of cancer cells in parallel with their non-cancerous counterparts because of the multiple factors involved in redox regulation and the stress response.

Furthermore, in order to utilize CAP as a promising tool for cancer therapy, investigations into identifying the distinct cellular and molecular responses between cancer and non-cancerous cells and the associated cellular mechanisms are still needed. By choosing the appropriate plasma source and dosage, plasmas can be selective and thus have an apoptotic effect on cancer cells without damaging healthy cells [1,17,35]. This paper examines the cellular effect of the RONS produced by atmospheric pressure-pulsed plasma jets on non-cancerous and melanoma cells. In order to examine the CAP-induced differential responses, we used melanoma (Hs 895.T) and non-cancerous skin fibroblast cells (Hs 895.Sk) (counterparts; both cell types are fibroblasts and were isolated from the same patient).

It is generally known that CAP can generate RONS in the gas phase [11]. Many studies have shown that after CAP irradiation, the accumulation of RONS increased in the CAP-irradiated medium [7,8,11,36]. Such findings suggest that CAP-induced RONS in the “gas and liquid phase” could result in intracellular RONS generation and apoptotic cell death [7,11,12]. However, it remains unclear how many RONS can enter the cells and interfere with intracellular signaling pathways [7]. Moreover, many studies reported CAP-induced intracellular reactive oxygen species (ROS) generation by using the 2′,7′-dichlorodihydrofluorescein diacetate (DCFH-DA) technique to estimate the total oxidative species [6,13,37,38,39]. However, DCFH-DA cannot reliably and selectively distinguish specific RONS [6]. Because the effects of an individual species of RONS in cells after plasma exposure remain uncertain, it is still needed to measure more specific intracellular RONS induced by CAP, such as NO, •OH, and H_2_O_2_, entering malignant and non-malignant cells of the same cell type from the gas and liquid phase. To detect highly reactive species, we used fluorescence probes, such as 4,5-diaminofluorescein diacetate (DAF-2DA, for intracellular NO) and 3’-(*p*-aminophenyl) fluorescein (APF, for intracellular •OH, peroxynitrite anion (ONOO^−^), and hypochlorite anion (ClO^−^)). To establish whether the overproduction of H_2_O_2_ in cancer cells enhances selectivity, the quantitative concentrations of H_2_O_2_ after CAP treatment in both non-cancerous and cancer cells were determined by an H_2_O_2_ colorimetric assay and intracellular H_2_O_2_ detection assay.

In this study, we examined whether CAP could selectively induce cell death in the melanoma cell line, while minimizing cytotoxicity in the non-cancerous fibroblast cell line. We also studied whether the selectivity of CAP treatment could be explained by the difference in the production of intracellular RONS between cancer cells and non-cancerous cells.

## 2. Results

### 2.1. Plasma Source Characterization

CAP jets are mostly topical and are particularly suitable for localized, relatively easily accessible cancers, such as skin cancer. Figure 1a shows a photograph of the plasma plume and the schematic of the experimental setup of the CAP jet driven by relatively moderate unipolar high-voltage square pulses with a repetition rate of several tens of kHz (FT-Lab PDS 4000). The system consists of two electrodes, in which one electrode is insulated from a pencil-shaped ring-grounded electrode by dielectric materials (tefron and polyetheretherketone (PEEK)). Thus, the CAP jet employed in this study can be regarded as a dielectric barrier discharge (DBD) jet. The CAP source has been described in detail in a previous study [34]. The typical operating conditions are an applied voltage of 1.7 kV_pp_, repetition frequency of 50 kHz, gas flow rate of 2 L/min, and duty ratio of 10%, unless otherwise stated. A relatively moderate voltage applied to the electrode allows the cold plasma to interact with organic materials without causing thermal/electric damage to the bio-surface.

The distance from the plasma exit hole to the cell surface was 10 mm. The waveforms of the voltage and current were measured using a real-time digital oscilloscope (LeCroy WS44XS-A) via a high-voltage probe (Tektronix P5100) and current monitor (Pearson 4100). To investigate the temporal behavior of the optical emission from the CAP jet, a photosensor amplifier (Hamamatsu C6386-01) was used to observe the wavelength-integrated plasma emission. Figure 1b presents the waveforms of the voltage and total current (upper row of the figure) and temporal evolutions of the wavelength-integrated optical emission intensities at the nozzle exit (bottom row of the figure). The discharges occur in the rising (primary discharge) and falling (secondary discharge) periods of the voltage waveform. Figure 1c illustrates the optical emission spectra recorded in the wavelength range of 200 to 900 nm using a fiberoptic spectrometer (Ocean Optics USB-2000) at an applied voltage of 1.9 kV_pp_. The optical probe was placed at a distance of 1.0 cm in front of the plasma jet nozzle. The data were collected with an integration time of 100 ms. The richness of highly reactive species such as nitrogen (N_2_), nitric oxide (NO), nitrogen cation (N_2_^+^), atomic oxygen (O), atomic hydrogen (H), and hydroxyl radicals (•OH) in our jet source could provide efficient applications in various plasma treatments.

Assuming that the structure of the dielectric barrier discharge (DBD) is of parallel plate type with one electrode covered by a dielectric and homogeneous barrier, the equivalent circuit model was introduced [40,41]. The equivalent capacitances of the dielectric barrier and the discharge gap, *C_d_* and *C_g_*, respectively, can be obtained from the Lissajous figure, which represents the relation between the transported charge and the applied voltage. A typical Q–V Lissajous figure of the plasma jet is shown in Figure 2a. It was obtained for an applied voltage amplitude of 1.5 kV_rms_, 50 kHz sinusoidal signal frequency, and gas flow of 2 L/min. The values of *C_d_* and *C_g_* were estimated as 16.41 pF and 8.69 pF, respectively. Utilizing the measured waveforms of *U_a_* (t) (applied voltage to the DBD cell) and *I_t_* (t) (total current through the DBD cell), the equivalent circuit model calculates the temporal evolutions of the voltage across the dielectric *U_d_* (t) (not shown), the voltage across the discharge gap *U_g_* (t), and the voltage across the discharge gap *I_g_* (t). The instantaneous power consumed by the plasma discharge in the discharge gap was calculated from equation *P_g_* (t) = *U_g_* (t) × *I_g_* (t): (Figure 2b). It should be noted that the secondary discharge also contributes to the power dissipation. The dissipated energy per cycle was found to be 17 μJ, and the specific energy input (SEI) [42] was 25.5 J/L.

### 2.2. Ozone (O_3_) Detection in Gas Phase and Generation of H_2_O_2_ and Nitrite (NO_2_^−^) in Liquid

Because O_3_ is one of the active species formed by CAP [43], an O_3_ detector was used to measure the O_3_ generation. The average O_3_ concentration increased from 0.265 to 8.952 ppm with increasing applied voltage (1.4 kV_pp_ → 2.5 kV_pp_) (Figure 3a). CAP-generated O_3_ has been paid much attention due to its strong oxidation and long lifetime in biomedical applications. It can reach the air-buffer solution interface during treatment on cells and produce secondary oxidants, such as •OH, O, and superoxide anion formed in the decomposition of O_3_ in liquid [43,44]. Figure 3b shows the measured concentration of H_2_O_2_ and NO_2_^−^ in the CAP-treated Hanks’ balanced salt solution (HBSS) without cells for different applied voltages. The H_2_O_2_ and NO_2_^−^ concentration increased with increasing applied voltage, which implies that the O_3_ concentration correlates with the concentrations of RONS in general.

### 2.3. Effects of Plasma Treatment on Oxidative Stress-Induced Mitochondrial Calcium (Ca^2+^) Increases

Ca^2+^ is a second messenger that mediates many physiological processes, such as differentiation, apoptosis, and oxidative stress. Ca^2+^ signaling homeostasis plays an important role in maintaining cellular processes [45]. Mitochondrial Ca^2+^ overload is one of the pro-apoptotic ways to induce the swelling of mitochondria, with perturbation or rupture of the outer membrane, resulting in the release of mitochondrial apoptotic factors into the cytosol [46]. In our previous studies [33,34,47], we investigated various cellular responses by plasma treatment in cancer cells (mostly the A549 lung cancer cell line, which is one of the most widely used for cancer research) [48]. We observed that this cell line was highly sensitive to plasma exposure (the overproduction of RONS and apoptotic behavior) [33,34,47]. In order to further investigate the effects of plasma treatment on oxidative stress-induced mitochondrial Ca^2+^ increases, we used Rhod-2 AM to analyze the mitochondrial Ca^2+^ levels with a microscope. To confirm that Rhod-2 AM can be used to show the level of mitochondrial Ca^2+^, we treated the lung cancer cells (A549) with the mitochondrial reactive dye MitoTracker Green and co-stained with Rhod-2 AM. Schneider et al. revealed that the CAP treatment of malignant melanoma cells causes a cytoplasmic Ca^2+^ increase derived from intracellular stores, and claimed that the induction of senescence depends on this Ca^2+^ influx [49]. Their previous study showed dose-dependent effects on those melanoma cells, in which a 2 min CAP treatment caused apoptosis, and a 1 min CAP treatment caused senescence [50]. They claimed that the direct and the indirect CAP treatment had differences in results with a few seconds time delay for responses, and that a longer CAP treatment led to a stronger increase in intracellular Ca^2+^ [49]. As shown in Figure 4, the results indicate that, after CAP treatment, mitochondrial Ca^2+^ levels increase with increasing applied voltage. The overwhelming Ca^2+^ levels within the mitochondria efficiently induce and trigger cell death [51]. Although the involved chemical mechanisms still need to be identified for various cell lines, our direct plasma exposure could affect Ca^2+^ homeostasis in lung cancer cells (A549) in a dose-dependent manner and may correlate with apoptotic cell death (observed in our previous study [34]).

### 2.4. Generation of Extracellular and Intracellular H_2_O_2_

Melanoma (Hs 895.T) and non-cancerous skin fibroblast cells (Hs 895.Sk and WS1) were used for the comparative study. Figure 5a shows the results of the H_2_O_2_ quantitation assay performed on serum-free HBSS (3 mm layer)-containing cells. The extracellular H_2_O_2_ concentration increased with increasing treatment time. To quantify the H_2_O_2_ levels in living cells, we used a unique cell-permeable sensor that generates a fluorescent product after reaction with H_2_O_2_. There were no significant differences in the levels of extracellular H_2_O_2_ between the non-cancerous and cancer cells. However, the intracellular H_2_O_2_ levels significantly increased in the cancer cells (Figure 5d) compared to the non-cancerous cells (Figure 5b,c).

### 2.5. Intracellular Generation of NO and •OH

The intracellular generation of NO and •OH after CAP treatment was detected using DAF-2DA and APF, two cell-permeable specialized fluorescence probes for the selective detection of highly reactive species. To investigate the selective impact of CAP, we analyzed the effects of RONS production on melanoma cells and non-cancerous cells. ROS include O_2_^−^, H_2_O_2_, and •OH, and reactive nitrogen species mainly include NO, NO_2_^−^, nitrate (NO_3_^−^), and dinitrogen trioxide (N_2_O_3_), all of which have inherent chemical properties that confer reactivity to different biological targets [52]. The excessive production of these species causes oxidative stress and plays an important role in cell death [24]. Figure 6 shows the fluorescence images of the intracellular •OH and NO production and the brightfield images. Firstly, it was observed that most of the cancer cells (Hs 895.T) morphed from spread to round cell shapes after CAP exposure, suggesting that they were affected to a greater degree than the non-cancerous cells (Hs 895.Sk, WS1) under the same CAP treatment condition. Moreover, the level of •OH generation in the CAP-treated cells significantly increased in the cancer cells. In non-cancerous cells, they were also increased compared with the control, but much less than in cancerous cells (Figure 6a). The same pattern was seen in the intracellular NO production (Figure 6b).

### 2.6. Measurement of Extracellular H_2_O_2_ and Cell Viability

In order to examine the influence of direct CAP treatment on the liquids containing cancer and non-cancerous cells, the concentrations of H_2_O_2_ were measured. Melanoma (Hs 895.T) and non-cancerous skin fibroblast cells (WS1) were used for the comparative experiments. As shown in Figure 7a, the concentration of the extracellular H_2_O_2_ in HBSS increased with increasing treatment time. Although there were no differences in the amount of extracellular H_2_O_2_ production immediately after both types of cells were treated with CAP, the H_2_O_2_ concentration in the HBSS containing the cancer cells was lower than that in non-cancerous cells at 1 h after treatment. These results suggest that the H_2_O_2_ consumption rate was faster in CAP-treated HBSS containing cancer cells than in non-cancerous cells. To visualize the number of cells remaining after CAP treatment at different voltages for 90 s, both cell types were incubated for 1 h at 37 °C, and then washed with phosphate-buffered saline (PBS) and incubated for 24 h in fresh medium with 0.1% fetal bovine serum (FBS). As seen in the brightfield images (Figure 7b), the numbers of cancer cells significantly reduced with increasing applied voltage (1.7 kV_pp_ → 1.8 kV_pp_) compared with the gas-control and non-cancerous cells. Cell viability was assessed by 3-(4,5-dimethylthiazol-2-yl)-5-(3-carboxymethoxyphenyl)-2-(4-sulfophenyl)-2H-tetrazolium, inner salt (MTS) assay. The viability of the cancer cells dramatically decreased with the increase in the voltage, which indicates fatal damage to the cell by CAP (Figure 7c).

## 3. Discussion

In our previous study [34], the cancer cells were more sensitive to plasma-induced RONS than non-cancerous cells, and plasma treatment induced apoptotic cellular responses, mostly in the cancer cells. For decades, many studies reported that the significant rise of intracellular ROS is the most common response of CAP-treated cells [6,26]. The significance of the increased intracellular ROS in the anti-cancer capacity of CAP has been widely verified through the observation that CAP cannot suppress the growth of cancer cells if those cells have been pre-treated with ROS scavengers such as N-Acetyl-Cysteine (NAC), D-mannitol, rotenone, and apocynin [6,12,39,47]. There remains debate regarding the selectivity of CAP at preferentially inducing cell death in tumor cells over healthy cells [8]. Discrepancies between reports can be attributed to the fact that the anticancer capacity of CAP treatment depends on several factors, such as the size of the wells in which cells are seeded, the volume of the treated liquid, the liquid composition, the gap between the CAP source and the liquid, the gas flow rate, the gas admixture, and the CAP device itself [7]. There is an optimal regime for direct CAP treatment where selectivity can be achieved, above which the oxidative stress becomes too overwhelming, even for non-cancerous cells [8]. Moreover, the cell type, cancer type, and culture conditions strongly influence plasma treatment and hence need to be considered when selectivity is determined [7,8]. Thus, in this study, we investigated the effect of particularly suitable direct CAP treatments on melanoma and non-cancerous skin fibroblast cells (counterparts, isolated from the same patient, cultured in the same medium, and treated in the same HBSS buffer under the same experimental conditions).

There were no significant differences in the levels of extracellular H_2_O_2_ between the non-cancerous and the cancer cells (Figure 5a). However, the intracellular H_2_O_2_ levels significantly increased in the cancer cells (Figure 5d) compared to the non-cancerous cells (Figure 5b,c). This indicates that cancer cells are more sensitive to the effects of CAP on the increase in the intracellular H_2_O_2_ level than non-cancerous cells. Although the generation of extracellular H_2_O_2_ is similar in both cancer and non-cancerous cells, the remarkable increase in intracellular production only occurs in cancer cells. This suggests that the function of the membrane channels of cancer cells may differ from that in non-cancerous cells to varying extents.

The intracellular generation of specific NO and •OH after CAP treatment was measured using cell-permeable fluorescence probes that enable the selective detection of highly reactive species. The level of •OH generation in the CAP-treated cells significantly increased in the cancer cells. In the non-cancerous cells, they were also increased compared with the control, but much less than in the cancerous cells (Figure 6a). In the direct plasma treatment, there is the possibility that short-lived species (such as hydroxyl radicals, singlet oxygen, and atomic oxygen) could contribute to cellular reactions. Although it is still hard to distinguish between species stemming from primary reactions and those created in secondary or tertiary reactions in the target, it is important to determine the reactivity of short-lived species to understand plasma chemistry [53]. Among those species, the hydroxyl radical (•OH) possesses extremely high reactivity, and numerous biomolecules are vulnerable to oxidative damage from hydroxyl radicals [54,55]. Given their high reactivity, the overproduction of OH after plasma treatment may be responsible for cellular disorders and cytotoxic effects that contribute to oxidative damage. As shown in Figure 6b, the same pattern appeared for intracellular NO production. Nitric oxide (NO) is a lipophilic, highly diffusible, and short-lived physiological messenger that regulates a variety of important physiological responses including cell migration, immune response, and apoptosis [56]. The biological effects of nitric oxide are dependent upon myriad factors such as the formation of the molecule, its metabolism, types of nitric oxide synthases, and concentration of nitric oxide [57]. High concentrations of NO appear to be toxic for cancer cells—causing cytostasis and apoptosis—whereas low concentrations are able to induce the activation of cancer-promoting pathways [58]. The CAP-generated excessive NO concentration in cancer cells may overwhelm the system and switch the NO effect toward tumor suppression [7]. CAP-related RONS are superoxide anions (O_2_^•−^), hydrogen peroxide (H_2_O_2_), hydroxyl radicals (•OH), nitric oxide (•NO), nitrogen dioxide (•NO_2_), PON (ONOO^−^), nitrite (NO_2_^−^), nitrate (NO_3_^−^), and singlet oxygen (^1^O_2_) [59]. Alimohammadi et al. found CAP treatment to increase endogenous NO production and lipid peroxidation, both associated with increased cell death in B16 tumor cells [60]. Chen et al. demonstrated that micro-sized CAP generates short- and long-lived species and radicals (i.e., hydroxyl radical (•OH), hydrogen peroxide (H_2_O_2_), and nitrite (NO_2_^−^)) with increasing tumor cell death in a dose-dependent manner [61]. As shown in Figure 5 and Figure 6, the melanoma cells responded more strongly than homologous non-cancerous cells to the CAP treatment, with a greater rise in H_2_O_2_, •OH, and NO. The excess production of RONS likely leads to either regulatory or cytotoxic effects selectively on cancer cells.

CAP initiates the dissociation of H_2_O molecules to form H_2_O_2_ as RONS in the aqueous medium, and H_2_O_2_ acts as a key intermediary in cell injury and death [29]. Even if H_2_O_2_ may activate the inflammatory responses, there is convincing evidence that the H_2_O_2_-generating system might be an efficient way of killing cancer cells in particular [62]. Moreover, the high levels of H_2_O_2_ seem almost incompatible with cell survival, and cancer cells seem more susceptible to H_2_O_2_-induced cell death than non-cancerous cells [32]. AQPs are H_2_O channel proteins that play a major role in transcellular and transepithelial H_2_O movement [63]. H_2_O_2_ is a soluble lipid and a strong oxidizing agent suggested to diffuse across cell membranes by some AQPs [62]. In general, tumor cells overexpress AQPs on their cytoplasmic membranes compared to homologous non-cancerous tissues [1,64]. A model based on the expression of AQPs was proposed to explain a correlation between RONS overproduction in CAP-stimulated Dulbecco’s modified Eagle medium (DMEM) and the selective effect of CAP on cancer cells [1]. Melanoma cells responded to CAP treatment with a greater rise in RONS and a higher consumption rate of H_2_O_2_ than homologous non-cancerous cells, resulting in a reduction of cell viability (Figure 7). These results support the model that the function of the H_2_O channels in cancer cells causes the selectivity of CAP treatment toward cancer cells. There are some reports supporting this model [65,66]. Yan et al. demonstrated that the anti-glioblastoma capacity of the plasma-stimulated medium can be significantly inhibited by adding AgNO_3_, which can block the water channels of aquaporins [65]. Bauer also demonstrated that apoptosis induction after CAP treatment was completely prevented in the presence of the aquaporin inhibitor Ag^+^ [66]. In plasma-cancer therapy toward selectivity, this hypothesis needs to be tested experimentally with inhibition and/or overexpression of aquaporins in order to establish its validity in further study.

## 4. Materials and Methods

### 4.1. Cell Culture

The Hs pair: non-cancerous skin fibroblast cells Hs 895.Sk(CRL-7636) and melanoma cells Hs 895.T (CRL-7637) and the human fibroblast WS1 cell line (CRL-1502) were obtained from the American Type Culture Collection (ATCC, Manassas, VA, USA). The lung adenocarcinoma cell line (A549) was provided by Professor Sun-Hee Leem (Dong-A University, Korea). The cells were grown in DMEM with 10% FBS and supplemented with antibiotics and glutamine.

### 4.2. CAP Treatment

A typical operating condition of the pulsed CAP jet was an applied voltage of 1.7 kV_pp_, a repetition frequency of 50 kHz, a gas flow rate of 2 L/min, and a duty ratio of 10% unless otherwise stated. Prior to CAP treatment, the medium from each chamber was almost completely removed, and a small amount of serum-free HBSS (a few hundred microliters) was left to keep cells wet during the treatment. The cells were exposed to CAP (and/or gas flow only) for 10–90 s on designated points per dish.

### 4.3. Measurement of O_3_ in the Gas Phase

The O_3_ monitor used in the present study (Model 202, 2B Technologies) is based on the well-established technique of the absorption of ultraviolet light at 254 nm. The Teflon tube was placed at a distance of 3 mm from the nozzle exit and connected from the O_3_ monitor intake to the plasma plume to directly record the plasma concentration of O_3_.

### 4.4. Measurement of H_2_O_2_ and NO_2_^−^ in Liquid

The H_2_O_2_ concentration was determined using an H_2_O_2_ colorimetric detection kit (Enzo Life Sciences). The dish plate (diameter, 20 mm) without cells was irradiated with plasma for 3 min, and samples of the liquids were taken from the dish after CAP treatment and dispensed into a 96-well plate to perform the Griess assay (and/or H_2_O_2_ detection assay). The Griess diazotization reaction is a two-step reaction in which nitrite is formed by the spontaneous oxidation of NO under physiological conditions. Sulfanilic acid is quantitatively converted to a diazonium salt by reaction with nitrite in an acid solution. The diazonium salt is then coupled to *N*-(1-naphthyl) ethylenediamine, forming an azo dye that can be spectrophotometrically quantitated based on its absorbance at 548 nm [67]. The absorbance at 548 nm (H_2_O_2_: 550 nm) was measured using a VersaMax microplate reader (Molecular Devices) after incubation for 30 min. In order to quantify the concentrations, a calibration curve was prepared using the standard sodium nitrite solutions (Molecular Probes) and standard H_2_O_2_ solutions (Enzo Life Sciences).

### 4.5. Measurement of Extracellular and Intracellular H_2_O_2_

The extracellular H_2_O_2_ concentration was determined using an H_2_O_2_ colorimetric detection kit (Enzo Life Sciences). A calibration curve was prepared using the H_2_O_2_ standard solutions. The dish plates (diameter, 20 mm) with cells (10^4^ cells) were treated with plasma, and after plasma treatment (60 and 90 s), samples of the liquids from each dish were dispensed into a 96-well plate to perform the assay. The absorbance at 550 nm was measured using a VersaMax microplate reader (Molecular Devices) following incubation (30 min). To quantify the H_2_O_2_ levels in living cells, we used the intracellular H_2_O_2_ assay kit (Sigma-Aldrich), which provides a unique cell-permeable sensor that generates a fluorescent product after reaction with H_2_O_2_, according to the manufacturer’s instructions. The cells were exposed to CAP (and/or gas flow only) for 90 s and incubated for 10 min with the assay solution.

### 4.6. Measurement of Intracellular RONS

APF has much more limited reactivity and higher resistance to light-induced oxidation; the fluorescein derivative is non-fluorescent until it reacts with •OH or ONOO^−^ [68]. In order to examine more specific reactive species, we used APF (10 μM). Likewise, the cells were loaded with the fluorescent cell-permeable NO-specific probe (DAF-2DA; 10 μM). The fluorescence was measured with excitation and emission wavelengths set at 488 and 520 nm, respectively. Cells in dishes (diameter, 60 mm; Corning) were pretreated with 10 μM of assay reagent for 5 min at 37 °C in the dark. Then, the cells were exposed to CAP (and/or gas flow only) for 10 s on nine points per dish and incubated for 5 min. The intracellular ROS production was observed on the marked points. After CAP treatment, the cells were washed with PBS. Fluorescence-activated cells were detected using a fluorescence microscope (Nikon) and quantified by measuring pixel intensity with an image analysis program (i-Solution, IMT i-solution, Inc., Canada).

### 4.7. MTS Assay

The cell viability was assessed by the MTS assay with the use of a kit (Promega) according to the manufacturer’s instructions. The MTS tetrazolium compound is bio-reduced by cells into a colored formazan product that is soluble in a tissue culture medium. This conversion is presumably accomplished by the NADPH (nicotinamide adenine dinucleotide phosphate; reduced form) or NADH (nicotinamide adenine dinucleotide; reduced form) produced by the dehydrogenase enzymes in metabolically active cells [69]. Assays were performed by adding a small amount of the solution reagent directly to culture wells, incubating for 2 h, and then recording the absorbance at 490 nm with the Victor3 spectrophotometer (Perkin-Elmer, CT, USA).

### 4.8. Detection of Mitochondrial Ca^2+^ Accumulation

Changes in the mitochondrial Ca^2+^ concentration were recorded using Rhod-2 AM. The cells were exposed to CAP (and/or gas flow only) for 10 s on nine points per dish. H_2_O_2_ (500 μM)-treated cells were used as the positive control, and gas-only-treated cells were used as the negative control. The cells were loaded with 5 μM Rhod-2 AM and 0.02% Pluronic F127 for 30 min at 37 °C. The cells were also loaded with 300 nM MitoTracker Green FM (MTG) to detect the localization of mitochondria.

### 4.9. Statistical Analysis

All data are reported as mean ± standard deviation (SD). The statistical significance of the difference between groups was analyzed by a two-tailed unpaired or paired Student’s *t*-test or by one-way analysis of variance (ANOVA) with Tukey’s post hoc test in instances of multiple comparisons (Prism, version 9; GraphPad Software, Inc., San Diego, CA, USA). * *p* < 0.05 was considered statistically significant.

## 5. Conclusions

Well-defined and relatively moderate high-voltage square pulses were properly employed to power the CAP jet for the CAP treatment of cancer cells in vitro. The selectivity of CAP treatment was tested precisely with cancerous and non-cancerous cells from the same tissue, the same cell type, and cultured in the same medium. A significant increase in the intracellular RONS (such as NO, •OH, and H_2_O_2_) production occurred in the melanoma cells, but not in their non-cancerous counterparts. The non-cancerous cells seemed to have a higher tolerance for RONS produced by CAP compared to the cancer cells; based on this, it is possible to target cancer cells while leaving non-cancerous cells unharmed. The H_2_O_2_ consumption rate was faster in the CAP-treated HBSS containing cancer cells compared to non-cancerous cells, resulting in a reduction in cell viability. The concept that H_2_O_2_ is always harmful has been widely revised because it may be therapeutically useful by killing cancer cells selectively [24,32]. Differential sensitivities of non-cancerous skin and melanoma cancer cells to CAP-induced RONS can enable the applicability of CAP in anticancer therapy. Nevertheless, further research is required to better comprehend the molecular mechanisms involved in cancer and non-cancerous cells and to discover specific CAP operating conditions in order to enhance selectivity.

## Figures and Tables

**Figure 1 ijms-23-14092-f001:**
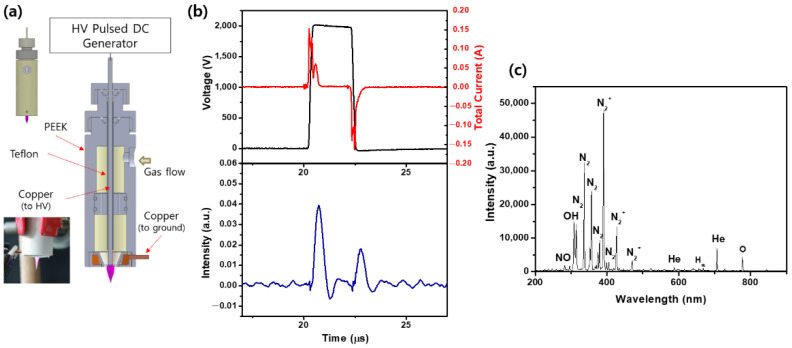
(**a**) Schematic of experimental setup and photograph of plasma plume. (**b**) Waveforms of the voltage (1.9 kV_pp_) and total current (the upper row of the figure) and temporal evolutions of the wavelength-integrated optical emission intensities at the nozzle exit (the bottom row of the figure). (**c**) Optical emission spectra.

**Figure 2 ijms-23-14092-f002:**
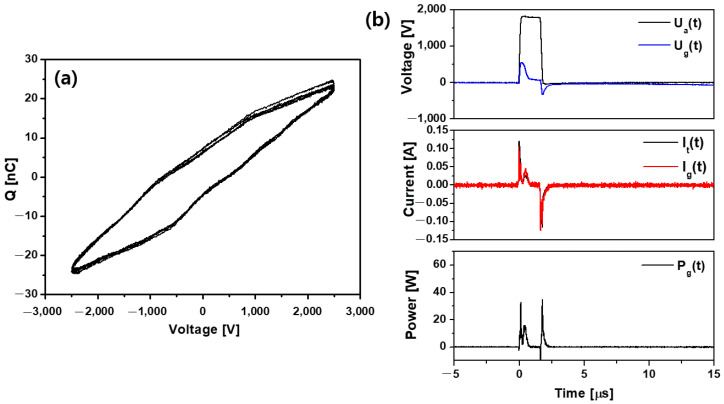
(**a**) Lissajous figures of the discharge. (**b**) Voltage and current waveforms and instantaneous power consumed by the discharge at the applied voltage of 1.7 kV_pp_, 50 kHz, duty ratio 8%. *U_a_* (t) is the applied voltage to the DBD cell, and *U_g_* (t) is the voltage across the discharge gap. *I_t_* (t) is the total current through the DBD cell, and *I_g_* (t) is the discharge (conduction) current in the gap. The instantaneous power is *P_g_* (t) = *U_g_* (t) × *I_g_* (t).

**Figure 3 ijms-23-14092-f003:**
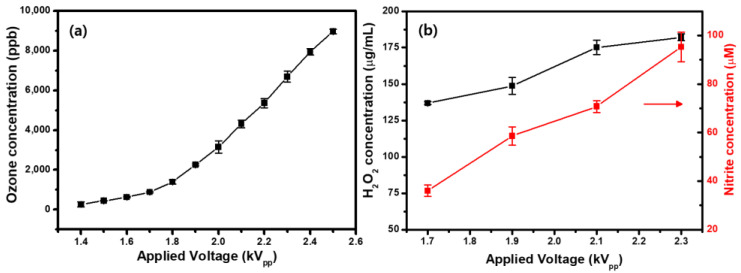
(**a**) Ozone (O_3_) concentration as a function of the applied voltage. (**b**) Generation of hydrogen peroxide (H_2_O_2_) and nitrite (NO_2_^−^) in cold atmospheric plasma (CAP)-treated Hanks’ balanced salt solution (HBSS) without cells. Each point represents the mean ± SD of three replicates.

**Figure 4 ijms-23-14092-f004:**
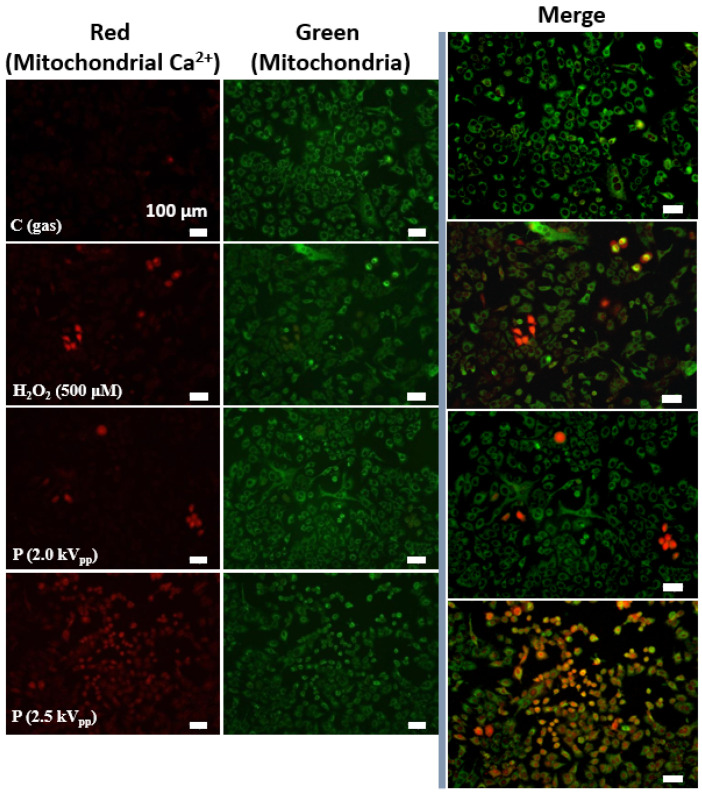
CAP induces the increased mitochondrial Ca^2+^ levels in dose-dependent manner in cancer cells. Images of the relative levels of mitochondrial calcium (Ca^2+^) as a function of the applied voltage in A549 lung cancer cells. Red images represent the detection of mitochondrial calcium (Rhod-2 AM), green images represent the detection of mitochondria (MitoTracker), and orange images indicate colocalization of the two dyes. Hydrogen peroxide (H_2_O_2_)-treated cells were used as the positive control, and gas-only-treated cells were used as the negative control. Scale bar indicates 100 μm.

**Figure 5 ijms-23-14092-f005:**
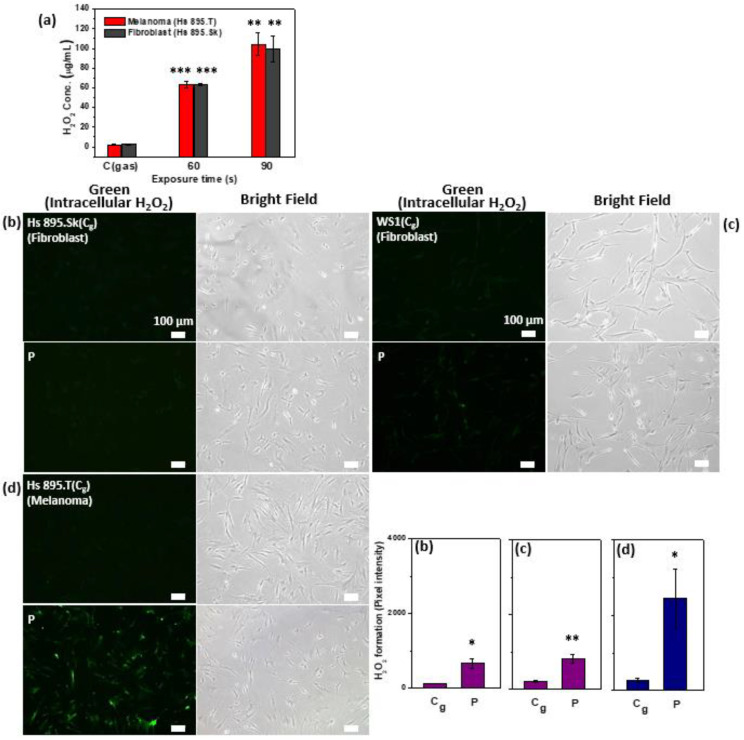
CAP induces the increased level of intracellular hydrogen peroxide (H_2_O_2_) in cancer cells. (**a**) Extracellular hydrogen peroxide (H_2_O_2_) concentration as a function of treatment time in the presence of cancer (Hs 895.T: melanoma) and non-cancerous (Hs 895.Sk: fibroblasts) cells. Fluorescence images and intensity graphs of intracellular H_2_O_2_ production and brightfield images in non-cancerous ((**b**) Hs 895.Sk, (**c**) WS1), and (**d**) melanoma (Hs 895.T) cells (upper row of figure, gas-treated control; bottom row of figure, CAP-treated cells). Green images represent the detection of intracellular H_2_O_2_. Each point represents the mean ± SD of three replicates. Scale bar indicates 100 μm. *, *p* < 0.05; **, *p* < 0.01; ***, *p* < 0.001. * *p* < 0.05 was considered statistically significant.

**Figure 6 ijms-23-14092-f006:**
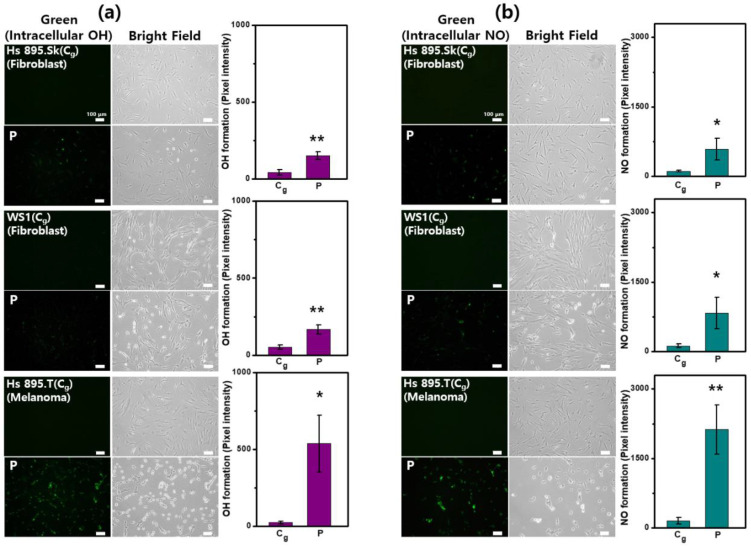
CAP induces the increased level of intracellular OH and NO in cancer cells. Fluorescence images (green: intracellular (**a**) •OH and (**b**) NO) and intensity graphs of intracellular RONS production, and brightfield images in gas-treated control and CAP-treated cells: (**a**) hydroxyl radical (•OH) (upper row, Hs 895.Sk (fibroblasts); middle row, WS1 (fibroblasts); bottom row, Hs 895.T (melanoma)) and (**b**) nitric oxide (NO) (upper row, Hs 895.SK (fibroblasts); middle row, WS1 (fibroblasts); bottom row, Hs 895.T (melanoma)). Each point represents the mean ± SD of three replicates. Scale bar indicates 100 μm. *, *p* < 0.05; **, *p* < 0.01. * *p* < 0.05 was considered statistically significant.

**Figure 7 ijms-23-14092-f007:**
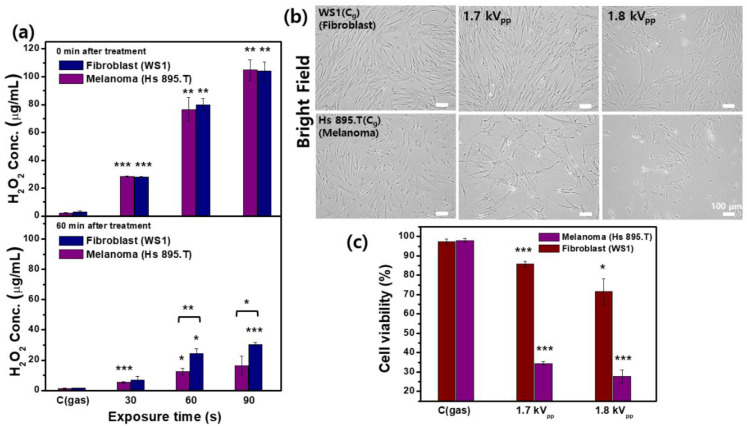
CAP induces the increased level of intracellular hydrogen peroxide (H_2_O_2_) and suppresses the cell viability in a dose-dependent manner in cancer cells. (**a**) Extracellular hydrogen peroxide (H_2_O_2_) concentration as a function of treatment time in the presence of cancer (Hs 895.T) and non-cancerous (WS1) cells (upper row, 0 min after treatment; bottom row, 60 min after treatment). (**b**) Brightfield images of cells (upper row, WS1 (fibroblasts); bottom row, Hs 895.T (melanoma)) and (**c**) cell viability measured by MTS assay at 24 h after CAP treatment (gas-treated, 1.7 and 1.8 kV_pp_). Each point represents the mean ± SD of three replicates. Scale bar indicates 100 μm. *, *p* < 0.05; **, *p* < 0.01; ***, *p* < 0.001. * *p* < 0.05 was considered statistically significant.

## Data Availability

All data are contained within the manuscript.

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
