# Peer review of "Differential Sensitivity of Melanoma Cells and Their Non-Cancerous Counterpart to Cold Atmospheric Plasma-Induced Reactive Oxygen and Nitrogen Species"

_ijms, 2022, doi:10.3390/ijms232214092_

Round 1

Reviewer 1 Report

This paper brings interesting results regarding the production of different RONS by CAP treatment and the differential consumption of them and consequent rise in intracellular RONS between cancerous and non-cancerous cells. However, we found major requirements in the experimental part to get your conclusions, as well as major text and figure corrections for better understanding of this work.

Experiments:

·         In this study, you compare the effects of plasma treatment between a cancerous melanoma cell line (Hs 895.T) and two corresponding non-cancerous counterparts (Hs 895.Sk and WS1). We missed consistency regarding the employment of this cell lines. For example, you compare Hs 895.T with Hs 895.Sk in Figure 5, but with WS1 in figure 7; you measure intracellular NO in the three cell lines in Figure 6 but you do not measure intracellular OH- in WS1, and in Figure 4 you use a completely different cancer cell line (A549). We suggest to compare Hs 895.T with at least 1 of the two non-cancerous cell lines for all the experiments.

·         Then, you claimed Hs 895.Sk and WS1 to be the corresponding counterparts of Hs 895.T cells. However, although both of them come from the skin of the same patient and are closely related with melanoma cells, they are fibroblasts (you described Hs 895.Sk cells as normal skin cells but if you check the ATCC they are described as fibroblasts). Melanoma cancerous cells are originated from melanocytes, for this reason we suggest to include non-malignant melanocytes in your studies.

·         At the beginning of the discussion (line 257) you claimed that the major cytotoxic effects induced by CAP are due to the production of RONS. In order to correctly correlate this production of RONS in external fluids, extra experiment by adding external inhibitors of RONS production and their repercussion in the increase of intracellular RONS and corresponding cytotoxic effects are required.

·         From line 275, you strongly suggest that the differences founded in the consumption of extracellular RONS and their corresponding differential increase of intracellular RONS and cytotoxic effects are a consequence of the higher expression of aquaporins in cancerous cells than in non-cancerous cells. First, to compare the expression of aquaporins between cancerous and non-cancerous cells. Then, to validate the paper of the aquaporins increasing H2O2 consumption in cancerous cells, knock-down/inhibition of aquaporins should decrease the consumption, intracellular increase and cytotoxic effects of CAP. On the other hand, overexpression of aquaporins in non-cancerous cells should have the contrary effect.

Writing:

·         The abstract does not include any introductory part explaining the background of the study neither their aims of the study, directly starting with materials and methods. You should include introduction and aims in the abstract.

·         We suggest to change the term “normal cell” to “non-cancerous cell” in all the text.

·         In line 16, suppress the parenthesis “(similar to 16
trypsin-mediated de-adhesion)” and from the rest of the test. We think it is not required.

·         In line 38, you mention the indirect treatment, but this is not described anywhere. You need to explain what does it consist.

·         In line 40, you discussed how RONS could be pro- or anti-tumorigenic, but this opposite effect depending of the dose are not well explained.

·         From line 46, deeper explanation about the paper of the different RONS produced by CAP are required.

·         We suggest to make more clear the aim of the study in the end of the introduction.

·         In Section 2.1. as well as in the material and methods, the feed gas (He according to the OES) is not described neither how OES is done. If the characterization comes from other paper, please explain it briefly and cite it.

·         In line 155 the measurement of ozone is not described. In addition, the role of ozone in CAP effects is not described and related with the experimental setup in any part of the text.

·         In line 218 you state that intracellular RONS are not increased in non-cancerous cells. However, they are significantly increased compared with the control, but much less than in cancerous cells.

·         In the discussion, you need major explanation about the paper of the different RONS produced by plasma and if H2O2 have a major role or not, based in your results.

·         In the discussion, the 2nd and 4th paragraph essentially discussed the same about aquaporins.

Figures:

·         We suggest to change the names of the cell lines by an easier name (like melanoma, fibroblasts, etc.) or code to identify them, as Hs 895.T and Hs 895.Sk have a really similar name and sometimes is confusing

·         In all the figures containing images, we suggest to indicate what is observed in each image on top of them. For example, in Figure 4, red images detect calcium, green images detect mitochondria, bright field images, etc. Same for each type of intracellular RONS detection.

·         In Figure 4, we suggest to provide the merged images between channels and also images with high magnification to better observation of the co-localization of both dyes,

·         In Figure 5, we suggest to increase the brightness of the images or provide with higher magnification, as they are not well observed.

Reviewer 2 Report

Malignancy treatment is one of the most important tasks of medicine today. Melanoma is one of the most dangerous malignant human tumors, often relapsing and metastasizing to almost all organs. The authors studied the interaction of cold atmospheric plasma induced nitrogen and oxygen with melanoma and normal skin cells, thus the article has the necessary scientific significance. The text is written clearly and interestingly, however, before the recommendation for publication, there are several comments for the authors of the article:

1. Apparently, V should be indicated instead of kV in Figure 1 (b).

2. Scale markers are hard to see on some images in Figures 4-7.

3. It is not entirely clear what p means in the figures 5-7. Perhaps it would be nice to clarify how p is calculated.

Round 2

Reviewer 1 Report

I I take into account you do not have enough time to do extra experiments. However, you claimed you focused on the cell line that is more susceptible to plasma exposure (WS1), but you do not use it in all the experiments (it is missing in Fig 5 and Fig 6a). I strongly encourage you to compare the effect employing the same non-cancerous cell line in all the experiments. Moreover, I suggest you to use the Hs 895.Sk cells, as they come from the same patient along with the melanoma cell line and you need to do less experiments (only Fig 7). In addition, if they are less sensitive you possibly are going to have greater differences with cancer cells.

In addition, although your experiment employing A549 cells provide interesting results, it elucidates a cell death mechanism (dependent of calcium efflux on mitochondria) that could vary between cancer cell lines. I recommend you to cite the study of Schneider C, et al. were this mechanism is observed in melanoma cells “Cold atmospheric plasma causes a calcium influx in melanoma cells triggering CAP-induced senescence”, however, this work suggest the induction of senescence, for this reason I strongly encourage you to validate your results employing melanoma cells.

Moreover, you correlate the differential consumption of H2O2 and consequent intracellular increase of this ROS with the overexpression of AQPs in cancer cells, however, you also observed a differential intracellular increase of other RONS like OH and NO that can not be explained by the AQPs hypothesis. More discussion about other possible mechanisms to observed this differences also in other species different that H2O2 is required.

Round 3

Reviewer 1 Report

I think now the work is OK